# Pharmacy Students’ Perspectives on Human Resource Management: An Examination of Knowledge and Attitudes

**DOI:** 10.3390/pharmacy12010027

**Published:** 2024-02-02

**Authors:** Georges Adunlin, Amy E. Broeseker, Jonathan L. Thigpen, Elizabeth A. Sheaffer, Marc Calhoun

**Affiliations:** Department of Pharmaceutical, Social and Administrative Sciences, McWhorter School of Pharmacy, Samford University, Birmingham, AL 35229, USA; aebroese@samford.edu (A.E.B.); jlthigpe@samford.edu (J.L.T.); esheaffe@samford.edu (E.A.S.); mcalhoun@samford.edu (M.C.)

**Keywords:** pharmacy, education, human resource management, knowledge, attitude, competency

## Abstract

(1) Background: This study aims to examine pharmacy students’ perceptions of their knowledge and competencies in human resource management (HRM) while also investigating their attitudes toward the educational content provided in a didactic HRM course. (2) Methods: A survey evaluating both course knowledge (pre and post) and attitudes was administered to students enrolled in an HRM class. Data were analyzed using descriptive statistics and measures of associations. (3) Results: All 98 course enrollees completed the survey (*N* = 98), revealing statistically significant knowledge growth across HRM topics from pre- to post-survey (*p* < 0.05). Notably, emotional intelligence, workforce diversity, conflict resolution, and recruitment strategies exhibited the most substantial increases. The expert panel session proved highly effective, with 71% reporting it as the most knowledge-enhancing activity. “Global and cultural effectiveness” emerged as the most valued competency, reflecting a positive overall attitude towards HRM. (4) Conclusions: HRM competency is one of the most fundamental skills for pharmacists, as many problems faced by pharmacy organizations and their solutions stem from the workforce. Pharmacy schools should therefore assess their curriculum to ensure that HRM is adequately addressed to meet accreditation standards and to prepare students to navigate HRM challenges in their workplaces post-graduation.

## 1. Introduction

Human resource management (HRM) is “the process of employing people, training them, compensating them, developing policies relating to them, and developing strategies to retain them” [1]. HRM includes all aspects of people management to meet an organization’s goals effectively. Over the years, the field of HRM has continually changed to reflect the ever-evolving workplace and is increasingly viewed as one of the most significant contributors to the success of an organization.

Pharmacists are one of the most accessible healthcare providers to the public [2]; hence, providing effective HRM training for pharmacists is critical to delivering high-quality healthcare services and fostering an efficient work environment. While HRM has been traditionally associated with the recruitment and compensation process, it is now increasingly recognized as a strategic function that aims to align an organization’s business strategy with its employees. All pharmacy organizations, irrespective of their size and structure, require effective human resource strategies for success. Cognizant of this, both the Accreditation Council for Pharmacy Education’s (ACPE) accreditation standards and guidelines and the American Association of Colleges of Pharmacy’s (AACP) Center for the Advancement of Pharmaceutical Education (CAPE) outcomes emphasize the importance of equipping pharmacy students with HRM skills [3,4]. Domain 2.2 of the CAPE Educational Outcomes outlines that, as part of essentials for practice and care under “Medication use systems management (Manager)”, students will learn to “manage patient healthcare needs using human, financial, technological, and physical resources to optimize the safety and efficacy of medication use systems.” Additionally, CAPE learning objective 2.2.4. specifies that pharmacy students should be able to “identify and utilize human, financial, and physical resources to optimize the medication use system” [5]. More recently, as an offshoot of the CAPE outcomes, the AACP Curriculum Outcomes and Entrustable Professional Activities (COEPA) 2022 sub-domain 2.6 emphasizes that pharmacists should “optimize patient healthcare outcomes using human, financial, technological, and physical resources to improve the safety, efficacy, and environmental impact of medication use systems” [5].

HRM skills are indispensable in the realm of pharmacy careers. Pharmacists must possess proficient HRM skills to facilitate seamless team collaboration, resolve conflicts effectively, and cultivate a positive workplace environment—all pivotal elements in ensuring optimal patient care. It is plausible to assert that HRM competency should be considered a cornerstone skill for pharmacists, as numerous challenges faced by pharmacy organizations, along with their corresponding solutions, often hinge upon workforce-related issues. This study aims to examine pharmacy students’ perceptions of their knowledge and competencies in HRM while also investigating their attitudes toward the educational content provided in a didactic HRM course. The outcomes of this study will provide helpful insights for schools and colleges of pharmacy, describing suggested HRM knowledge and competencies vital for the future success of pharmacy students in the workplace. These findings could serve as a blueprint for crafting HRM content within pharmacy curricula, ensuring its ongoing appropriateness and relevance. We hypothesized that targeted interventions and educational initiatives focusing on HRM within the pharmacy curriculum will significantly enhance students’ perceived knowledge and foster a positive attitude towards HRM practices.

## 2. Materials and Methods

### 2.1. Course Description

The “Human Resource Management for Pharmacists” course was a required 3-credit course that met twice per week for a total of three hours. The course was taught in the third professional year of the Doctor of Pharmacy (PharmD) program. The course catalog description indicated that “Human Resource Management (HRM) is designed to equip students with essential personnel management and leadership skills necessary for practice in various pharmacy settings” [6].

Prior to enrolling in the HRM course, students were required to complete a 3-credit “Financial Management” course in their second professional year [7]. That course instructed students on how to allocate resources to support an organization’s goals and maintain a balance between costs and revenue. Then, the HRM course provided a comprehensive foundation for all aspects of human resources planning, development, and administration. Students were expected to acquire the knowledge and skills to make informed human resource decisions in the healthcare environment. This included the ability to understand, identify, and solve human resource problems, as well as evaluate possible outcomes related to the concepts, principles, and theories of leadership and human resources.

Table 1 lists the topics and major assessments covered in the HRM course, excluding quizzes and exams. The course was delivered in a blended learning format that combined didactic lectures with active learning activities. The didactic lectures were delivered through formal presentations by the course instructors and guest speakers. The course also featured an alumni panel discussion with a diverse group of pharmacy leaders representing various practice settings. During this session, these leaders shared their experiences and engaged in discourse on optimal approaches to address prevalent human resource challenges in the pharmacy field.

To enhance students’ critical thinking in the course, active learning activities were integrated through group-based learning with authentic tasks, scaffolding, and individual reports. The students were organized into 17 groups, and each group worked on their own Lean project, ensuring a balanced and equitable workload for each group. The purpose of the Lean project was to provide students with the opportunity to apply the Lean methodology learned in class to a real-world project. Integrating Lean management into the course aims to equip graduates with the skills to streamline processes, eliminate waste, and adapt organizational structures. This practical knowledge enhances their ability to drive efficiency, contribute to strategic business improvements, and stand out in the competitive job market. Lean methodology is a rigorous method of optimizing the people, resources, effort, and energy of an organization toward creating increased value for the customer while minimizing waste [8] and, in the case of pharmacy, creating increased value for the patient. Lean manufacturing techniques provided the original framework for the application of Lean practices [8]. The students’ Lean projects covered a wide range of issues within and outside of the pharmacy field. The student groups were given autonomy in choosing their project topics. Examples of these student-produced Lean projects in the HRM course included “Improving vaccination efficiency within a community pharmacy setting”; “Creating and implementing a training program for new employees at a home infusion pharmacy”; “Improving vaccination process in a chain pharmacy”; “Standardizing vaccine procedures for safety and quality assurance”; “Color-coded bins in community pharmacy”; “Enhance the internal flow of a fresh food studio”; and “Reducing waste by streamlining the auto-refill process”.

In this course, the instructors also drew the connection between HRM and emotional intelligence. Emotional intelligence has been defined as “the intelligence to employ emotion and feeling toward guiding behavior, thoughts, and relationship with others, colleagues, supervisors, and clients and also to spend time to improve the outcomes” [9]. Emotional intelligence capacities include intrapersonal skills, interpersonal skills, adaptability, stress management, and general mood [10]. There is a high level of commonality between HRM, emotional intelligence capacities, and achievement in working life. While HRM is the function within an organization that is responsible for attracting, developing, and retaining employees, emotional intelligence is the ability to understand, manage, and use one’s own emotions and the emotions of others. Midway through the course, the students completed a one-week Introductory Pharmacy Practice Experience (IPPE), which is a required part of the PharmD program. IPPE is a supervised experiential learning opportunity that allows pharmacy students to apply the knowledge and skills they have learned in the classroom to a real-world setting.

The session on emotional intelligence occurred right before the IPPEs. Following this session, students were required to complete two self-reflections at two-time intervals: one immediately after the session and another after they had completed their IPPEs. The purpose of the self-reflection assignments was to enable students to connect the knowledge and skills they had acquired in the HRM course to their observations and experiences in their IPPEs, with a focus on emotional intelligence. Before their IPPEs, the students were asked to reflect on the following questions: (1a.) Identify the two emotional intelligence elements you feel you are engaging in very well and explain their impact on your personal and/or academic success thus far; and (1b.) How will you use what you have learned about emotional intelligence as you go into your IPPE? After their IPPEs, students were asked to reflect on the following questions: (2a.) In what ways did you use emotional intelligence during your IPPE? Describe one example or situation where you were aware of using it; and (2b.) How will you use what you have learned about emotional intelligence in your future career as a pharmacist or during the rest of your time in pharmacy school?

### 2.2. Data Collection Tool

A survey evaluating knowledge (pre and post) and attitudes was administered to students enrolled in the HRM course. The survey instrument comprised demographic questions, 13 statements evaluating students’ perceived knowledge, and three statements assessing their attitudes toward HRM. The statements on knowledge were developed based on the course topics and objectives. All the statements are detailed in subsequent sections of the manuscript. Additionally, the survey featured a question about students’ future plans to acquire HRM knowledge, along with their ratings of the class activities. To assess the survey instrument’s potential flaws, the survey was pilot-tested with a small sample of five students not enrolled in the course. The group provided a few recommendations on the survey’s length and wording that needed more clarity. The appropriate changes were made based on the feedback received before the survey was administered to the students in the course. Participation in the study was voluntary, and informed consent was obtained from all participants. Personal identifiers were not collected. The survey was administered online through Qualtrics© XM (Qualtrics, Provo, UT, USA) [11]. A copy of the survey instrument is available upon request from the corresponding author.

### 2.3. Ethical Considerations

This study was reviewed and approved by Samford University’s Institutional Review Board (Reference number: EXMT-P-23-S-1).

### 2.4. Statistical Analysis

Descriptive statistics, including frequencies, means, standard deviations, and ranges, were used to summarize the data. Paired t-tests were used to compare participants’ responses on Likert-scale statements at the beginning and end of the course. The data were analyzed using STATA statistical software, version 17.0 [12]. Statistical significance was defined as a *p*-value of less than 0.05.

## 3. Results

All 98 students enrolled in the HRM course completed the survey, representing a response rate of 100%. The demographic characteristics of the study participants are summarized in Table 2. The majority of participants were female (63.27%), while 36.73% were male. Regarding work experience, 53.06% reported having 4 to 7 years of any kind of work experience, with 23.47% having 1 to 3 years. In terms of pharmacy work experience, nearly half (47.96%) had 1 to 3 years, while 34.96% reported 4 or more years. A notable portion of participants (35.71%) expressed postgraduate plans in hospital pharmacy, followed closely by 33.67% in community pharmacy. The majority of participants (83.67%) did not have managerial or supervisory experience.

### 3.1. Knowledge and Attitudes

The first part of the survey assessed pharmacy students’ perceived knowledge of HRM topics before and after the course. Table 3 displays the mean pre- and post-course scores, along with standard deviations, for all 13 HRM topics. The statistical significance of the difference between pre- and post-course knowledge is indicated by the *p* value. All HRM topics show a significant improvement in knowledge post-course (* *p* < 0.01), highlighting the effectiveness of the course in enhancing participants’ understanding across a range of HRM areas.

A one-time survey assessing attitudes was administered towards the conclusion of the course. Table 4 presents students’ responses, represented as both counts and percentages, regarding the students’ attitudes toward HRM. The statements cover willingness to learn more about HRM, intent to stay updated after completing the course, and the perceived importance of HRM as a topic in pharmacy education. Most students (63.27%) strongly agreed or agreed that they were willing to learn more about HRM in pharmacy school. Most students (76.53%) also strongly agreed or agreed that they would keep updated about HRM after completing the course. The vast majority (82.65%) also strongly agreed or agreed that HRM is an important topic to learn in pharmacy school. The majority of students expressed positive attitudes toward these aspects, highlighting a favorable disposition toward the significance of HRM in their academic and professional pursuits.

Following the assessment of attitudes towards HRM, student participants were queried about their primary training preferences in HRM. A majority of students (53.06%) expressed a preference for attending conferences, workshops, and/or seminars. Conversely, a notable proportion of students favored receiving training in a classroom setting (32.65%), while a smaller cohort opted for self-directed learning through personal readings (14.29%).

### 3.2. Competency Rankings

At the end of the course, students were asked to rank the relative significance of eight HRM core competencies crucial for pharmacy graduates entering the workforce. These competencies were obtained from the Society for Human Resource Management competency model [13]. Figure 1 involved students ranking HRM competencies in terms of importance for pharmacy graduates entering the workforce. Consequently, the data are consolidated based on their rankings. The findings revealed that “Global and cultural effectiveness” emerged as the highest-ranked competency at 20.83%, trailed by “Communication” at 17.71%, and “Organization leadership and navigation” at 15.63%. Conversely, “Business acumen” garnered the lowest ranking at 6.25%.

### 3.3. Rating of Course Activities

Figure 2 shows the evaluation of activities and assignments employed to enhance student engagement in the course. The students had the freedom to either select or opt not to select the activities, as this question allowed for multiple choices, allowing them to choose all that applied. Among the assessed activities, the alumni panel discussion garnered the highest rating from students (29.22%). The second highest-rated activities were the guest lectures delivered by faculty members from both within and outside the pharmacy school. Conversely, the Lean project and group exercises were deemed the least engaging activities, each receiving a rating of 7.41%. The Lean project was introduced later in the semester, and feedback indicated a preference among students for its earlier introduction. The segmented completion approach also generated stress and impeded a comprehensive understanding. Therefore, the lower rating assigned to the Lean project could have primarily been attributed to the timing of it and its design rather than its utility.

## 4. Discussion

This study investigated pharmacy students’ self-reported knowledge of HRM and competencies, as well as their attitudes toward the education they received in a didactic HRM course. Overall, there was a significant increase in students’ perceived knowledge of all topics between the beginning and end of the HRM course. The students’ attitudes also indicated that they generally saw the importance of HRM. Moreover, students expressed a keen interest in pursuing additional informal training in HRM after graduation.

Global and cultural effectiveness (i.e., the ability to value and consider the perspectives and backgrounds of all parties) was ranked by students as the most important HRM competency. In pharmacy, global and cultural effectiveness encompasses the knowledge and skills that pharmacy students need to be successful in today’s interconnected and culturally diverse society and to fully engage in and act on issues of global significance [14,15]. Cultural effectiveness in pharmacy is considered one of the most important factors in improving the efficiency and quality of care in the United States’ increasingly diverse population. The growth of cultural effectiveness has led to the development of several other related concepts, such as cultural awareness, cultural humility, and cultural competence. Cultural effectiveness training in pharmacy education and workforce development has evolved from simple training with limited touchpoints to longitudinal and more sophisticated training that combines theoretical knowledge, practical skills, and hands-on experience [16,17,18]. Considering variations in activity rankings in the course, instructors could optimize HRM design by emphasizing real-world case studies, interactive simulations, and group discussions for heightened engagement. However, it is crucial to consider previous student input when designing these activities.

In contrast to numerous pharmacy administration courses [7,19,20], no prior studies have assessed the extent of HRM education in United States colleges and schools of pharmacy. However, the COVID-19 pandemic drove many healthcare organizations, including pharmacy organizations, to evaluate their HRM strategies. In this context, a few studies on HRM emerged [21,22]. For example, Adam and colleagues highlighted seven human resources issues that were addressed as part of a pharmacy department response to COVID-19. These issues include: “crisis management, internal communications, employee stress, reorganization of workspaces, reorganization of pharmacist workforce, telework, and schedule management” [21]. The authors anticipated that the changes made in human resources practices of pharmacy departments during the COVID-19 pandemic are likely to remain in place after the pandemic. Considering the heightened risk of infection and increased susceptibility to mental health challenges among pharmacists during the COVID-19 pandemic [23,24], the implemented HRM policies and strategies were primarily aimed at enhancing pharmacist workplace safety and well-being. Consequently, discussions and assessments within pharmacy programs increasingly focus on topics related to pharmacist workload, burnout, well-being, and their impact on patient safety [25,26,27].

Various challenges impede the comprehensive teaching of HRM in pharmacy programs. Notably, the constrained time allocated within the curriculum for HRM content is a significant hurdle. Both HRM and other social and administrative subjects have frequently taken a backseat in pharmacy curricula. This predicament is exacerbated by the recent wave of curriculum transformation, where social and administrative topics contend for time in a curriculum increasingly emphasizing basic science and clinical courses.

In the HRM course, students successfully applied Lean skills to identify and eliminate process inefficiencies, delivering enhanced value to customers with optimized resource utilization. The incorporation of Lean principles and tools elevated the course experience, providing students with an opportunity to address significant organizational or community challenges akin to those encountered in their future pharmacy practice. Involving students in the practical application of Lean skills offers a framework applicable across diverse scenarios necessitating process modifications. Key advantages for students engaging in a Lean project include the application and exploration of knowledge acquired in the course, the translation of theoretical understanding into practical implementation, and the cultivation and application of competencies such as teamwork, project management, critical thinking, problem-solving, and communication skills. Additionally, participation in Lean projects fosters an awareness of the importance of professional life, an entrepreneurial attitude, and an initiative spirit, among other valuable attributes. Despite a scarcity of published reports on the implementation of Lean methodology in pharmacy education, a recent study underscored its efficacy. Davis and colleagues described the application by third-year PharmD students of Lean skills to a Medicare insurance counseling or Seniors Health Insurance Information Program certificate elective course [28]. The authors highlighted Lean as an efficient catalyst for clinical transformation, showcasing how the application of Lean skills is feasible within a service-learning course [28]. They further contended that Lean offers a versatile skill set that can be introduced and utilized across various instructional settings, equipping student pharmacists with crucial knowledge and skills in quality improvement [28]. Another study described the implementation of Lean methodology into pharmacy residency programs at a community teaching hospital [29]. According to the study, the integration of Lean methodology into pharmacy residency programs resulted in an array of tangible and prospective advantages for the residency programs, the preceptors and residents, and the health system [29]. Notable areas of enhancement encompassed quality improvement processes, the expansion of leadership opportunities for residents, and the improvement of communication among program directors, preceptors, and residents [29].

Pharmacists are confronted with various aspects of personnel management, a prominent feature of pharmacy due to its significant involvement in business and economics. Additionally, the escalating shortage of clinical staff poses a significant hurdle in maintaining high-quality care. In this context, implementing effective HRM strategies may offer a solution to alleviate staff shortages and enhance patient satisfaction. The entrepreneurial and competitive nature of pharmacy practice underscores the importance of effectively managing the human resource component for successful and profitable operations. Within the community pharmacy setting, human resources present distinct challenges, impacting patient relations, productivity, and the overall image of the organization.

In essence, this article’s objective was not to extrapolate representative conclusions applicable to all pharmacy students but rather to elucidate how their perspectives on human resource content in pharmacy education could contribute to curriculum revision or reform. Nevertheless, it is imperative to acknowledge certain limitations in interpreting the study’s results. Firstly, the assessment instruments were conducted within a singular academic institution, potentially constraining its generalizability. Secondly, reliance on self-reported data introduces the possibility of recall and response bias [30]. Thirdly, correlation analysis was precluded due to the non-fulfillment of the normality assumption required for a normally distributed set of variables. Subsequent research is warranted to evaluate the incorporation of HRM topics across curricula in United States pharmacy programs, encompassing both presently taught subjects and the instructional methods employed for content delivery.

## 5. Conclusions

HRM deals with an organization’s most important resource, which is its human capital. HRM is essential to the smooth running of any organization, including pharmacy organizations. This study highlights pharmacy students’ perceived knowledge and attitudes toward HRM and provides evidence of the competencies that students believe should be included in the pharmacy curriculum. Enhancing human resource management skills among pharmacists, who serve as essential members of the healthcare team, will positively impact their work environment and their perceived value and professional image within healthcare organizations. Conversely, the inadequacy of training for managerial roles and the absence of preparatory education in human resource skills for pharmacy graduates may potentially place them at a disadvantage in the competitive healthcare sector. The findings presented in this study offer valuable insights for schools and colleges of pharmacy, enabling them to assess their curriculum and ensure that pharmacy students receive thorough preparation for managing human capital in professional settings.

## Figures and Tables

**Figure 1 pharmacy-12-00027-f001:**
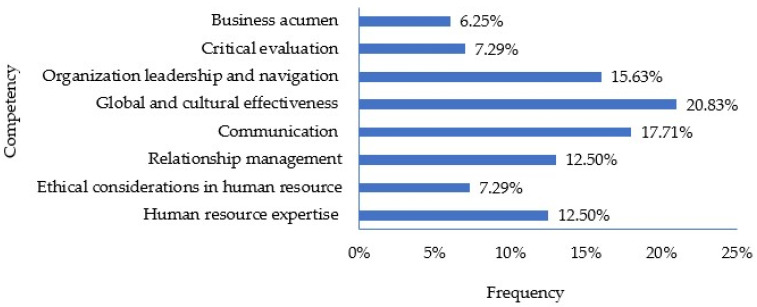
Ranking of human resource management competencies.

**Figure 2 pharmacy-12-00027-f002:**
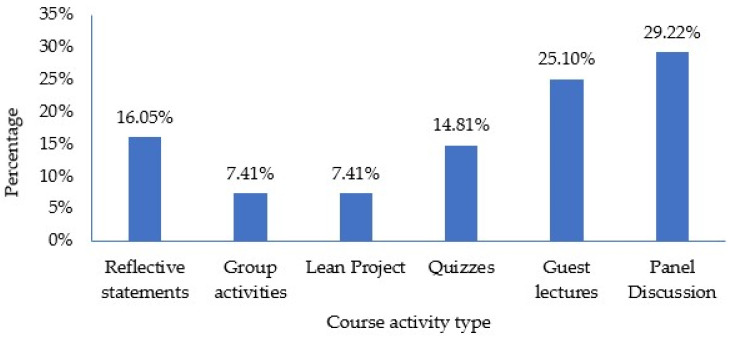
Rating of course activities.

**Table 1 pharmacy-12-00027-t001:** Topics and principal assessments (excluding quizzes and exams).

Module	Topic	Principal Assessments *
1	Organizational Structure and Behavior	-
2	Human Resource Strategy and Planning in Pharmacy	-
3	Human Resource Management Practices for Quality and Patient Safety	-
4	Workforce Diversity	-
5	Employment Law and Workplace Safety	-
6	Successful Recruitment and Hiring Strategies	-
7	Conflict and Negotiation	-
8	Job Design and Analysis	-
9	Covey Time Management and Burnout Prevention	-
10	Emotional Intelligence and Human Resources	Self-reflection on emotional intelligence; IPPE self-reflection
11	Effective Communication in Human Resource Management	
12–16	Lean Concepts in Human Resources	Lean project report; Lean project journal activities; Lean project presentation
17	Expert Panel “Lessons Learned in Human Resource Management”	-
18	Managing Pharmacy Human Resources through Emergency and Disaster	-
19	Performance Management Systems in Pharmacy Organizations	-
20	Addressing Human Resources in Pharmacy Residency	-

Note: Introductory Pharmacy Practice Experience (IPPE); * Excludes quizzes and exams.

**Table 2 pharmacy-12-00027-t002:** Sociodemographic characteristics of the participants (*N* = 98).

Variables	Frequency (%)
Gender	
Female	62 (63.27)
Male	36 (36.73)
Years of any kind of work experience	
Less than 1 year	3 (3.06)
1–3 years	23 (23.47)
4–7 years	52 (53.06)
8 years or more	20 (20.41)
Years of pharmacy work experience	
Less than 1 year	17 (17.35)
1–3 years	47 (47.96)
4 years or more	34 (34.96)
Managerial or supervisory experience	
Yes	16 (16.33)
No	82 (83.67)
Postgraduate plans	
Hospital pharmacy	35 (35.71)
Community pharmacy	33 (33.67)
Pharmaceutical industry	9 (9.18)
Others	7 (7.14)
Undecided	14 (14.29)

**Table 3 pharmacy-12-00027-t003:** Pre- and post-course perceived knowledge in human resource management (*N* = 98).

HRM Topics	Pre-Course Mean (SD)	Post-Course Mean (SD)	*p* Value ꬸ
Organizational Structure and Behavior	2.27 (0.95)	3.70 (0.84)	*
Strategic Human Resource Planning	2.16 (0.96)	3.75 (0.80)	*
Workforce Diversity	3.19 (1.07)	4.15 (0.80)	*
Human Resource Laws and Regulations	2.17 (0.95)	3.97 (0.83)	*
Job Design and Analysis	2.28 (0.99)	3.80 (0.87)	*
Recruitment and Hiring Strategies	2.59 (1.00)	4.09 (0.77)	*
Conflict Resolution in Human Resources	2.76 (0.96)	4.12 (0.82)	*
Human Resource Communications	2.47 (1.04)	3.81 (0.88)	*
Managing HR through Emergency and Disaster	1.98 (1.02)	3.58 (0.97)	*
Performance Management Systems	2.08 (1.11)	3.65 (0.94)	*
Emotional Intelligence	3.04 (1.09)	4.20 (0.82)	*
Burnout Prevention	2.77 (1.08)	3.97 (0.86)	*
Lean Concepts in Human Resources	1.46 (0.86)	3.59 (1.02)	*

Note: Perceived knowledge in HRM at the beginning and end of the course; ꬸ Significance in the difference between pre-and post-course knowledge; * *p* < 0.01.

**Table 4 pharmacy-12-00027-t004:** Attitudes toward human resource management (*N* = 98).

Statement	Response, *N* (%)				
	Strongly Agree	Agree	Neutral	Disagree	Strongly Disagree
I am willing to learn more about HRM in pharmacy school.	15 (15.31%)	47 (47.96%)	25 (25.51%)	6 (6.12%)	5 (5.10%)
2. I will keep updated about HRM after completing the course.	16 (16.33%)	59 (60.20%)	16 (16.33%)	6 (6.12%)	1 (1.02%)
3. HRM is an important topic to learn in the pharmacy school.	28 (28.57%)	53 (54.08%)	10 (10.20%)	4 (4.08%)	3 (3.06%)

Note: Attitude toward human resource management at the end of the course. Ratings were assigned on a scale of 1 to 5, where 1 signified “Strongly Disagree”, and 5 denoted ‘Strongly Agree”.

## Data Availability

Data are available upon request from the corresponding author.

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
