# Peer review of "Pharmacy Students’ Perspectives on Human Resource Management: An Examination of Knowledge and Attitudes"

_pharmacy, 2024, doi:10.3390/pharmacy12010027_

Round 1

Reviewer 1 Report

Comments and Suggestions for Authors

The manuscript aims to presents the design and evaluation of the course HRM for Pharmacist based on Lean management. The authors focused on introducing Lean methodology to students and enabling them to get to know its practical applications. The presented results showed an increase in perceived knowledge after the course as well as increased interest in HRM. The main strengths of the manuscript are as follows: innovative and robust course design, high number of course participants, evaluation of both knowledge, attitudes and course structure. Data presentation is clear and consistent, the article is well written. The conclusions are consistent with the presented evidence. The authors cited much literature, most of which is recent, but due to scarcity of articles concerning the use of lean methodology in pharmacy education, referring to some older articles is also understandable. The authors identified further knowledge gaps that could be addressed by further studies. I would raise some points that could be better addressed by authors:

1)     The authors addressed an important topic, i.e. the development of the HRM course that would teach the expected skills lacking in contemporary pharmacy curriculum. The research gap is well stated, however the primary hypothesis of the study could be better defined. 

2) Why did authors choose lean management method? What will be benefits for graduates of introducing specifically this methodology? I believe that since the course is about lean methodology, the description of lean could be moved from Methods section to Introduction.

3) The authors validated the effect of the course by measuring perceived knowledge in several domains. However, the perceived knowledge is still less informative and credible as compared to the acquired knowledge and skills proven by a relevant assessment method. Could authors comment on this? Maybe the results of quizzes could be included in manuscript?

4) Authors should include some more information regarding course design, i.e. total group size for common activities, group size for Lean projects. How many questions were included to test various aspects of perceived knowledge?

5) In table 4 authors presented students' interest in HRM following course. Did participant characteristics mentioned in table 2 (e.g. postgraduate plans or professional experience) influence these attitudes? This could be interesting for teachers working with curriculum design to better tailor educational offer for their students. 

6) The authors mentioned that Lean projects were important for students in translating theory into practice. Still, they were considered least engaging. What was the reason for that. This result should be discussed. 

7) Since various course activities were rated rather differently, i.e. raging from 7 to 29%, could authors make some recommendation on optimal HRM course design in Discussion? This could be useful to readers of Pharmacy.

8) Finally, most of cited literature is new and relevant, but I believe references 9 and 20 could be replaced with some newer articles. 

Reviewer 2 Report

Comments and Suggestions for Authors

Dear authors,

Thanks for giving me the chance to review this manuscript.

We understand that human resource management is an important “soft skill” for wide range of business activities, and business operators often complaining that university graduates are lacking in human skill when entering the workforce. Therefore, an innovation and research about incorporating this skill into the university curriculum is very useful. However, during the review of your manuscript, I observed some important issues that limited the contribution of the study and I hope you are able to address.

Major issues:

Reference: there is a distinct lack of reference in the introduction, and some of the reference are very poorly written, E.g. Reference 1 is not traceable, reference 3&4 are not complete, reference 5 is not legible, reference 8’s doi links to a different article, reference 9 is not traceable, reference 20’s doi do not exist, any many other errors in between. The refence list seems to be generated by reference managing software but the authors seem not able to use them properly and/or did not check the final version. Honest and accurate referencing should be taken seriously as it is the foundation of academic and scientific integrity.

The methodology is lacking in detail for reader to properly assess the reliability of data, and difficult for fellow researchers to reproduce the experiment. There is large amount of information given about course description, but it is poorly organised and confusing. However, when describing the data collection (noting a missing in study design section), it seems very vague and lack in essential details. For examples, in section 2.2., it stated that “a survey is administered to students enrolled in the course”. But Table 3 reported pre-course data and post-course date, meaning that there are at least 2 surveys. Then in section 3.1, the author describes there is “a survey assessing attitude”. It is not clear to a reader if this is a separate survey, the length of the survey, and how many survey in total was conducted and at what time/interval. Another point is regarding the lack of detail in survey design, e.g. it described that there are 6 statements to assess student perceived knowledge and attitude, but there is no information about what the 6 statements are, and the result only reported 3 statements in Table 4. In addition, the statements are written as non-neutral statements, and there is not explanation why such statement is chosen and its potential influence on the results. This is a well-studied area of social science research that should be acknowledged and keep into consideration when interpreting the findings.

The introduction will need some backgrounds and data from previous studies/reports regarding HRM in pharmacy profession. It should also include some backgrounds about existing information on HRM in pharmacy curriculum, as this topic although not necessarily as a standalone subject, often included as “soft skill” in professional practice type of subjects of many pharmacy curriculum.

In result section, Figure 1 and 2 supposed to illustrate the ranking/rating of various topics. However, I can only assume that the data represent only the top preference. It does not provide any information on the second rank, third rank or bottom ranks. In addition, it is also not clear how the ranking data is collected and analysed, e.g. are the students asked to choose one of the many option, or choose 3 top option, or rank every item?

The discussion seems to be very far divorced from the actual study and its findings (e.g. culture effectiveness), or at least from the reader point of view, no sufficient information is provided to make the connection. In addition, there appear to be a lot of discussion about the Lean concept, both in the methodology and in the discussion. However, looking at the topics outlined in Table 1, Lean concept is only one of the 19 topics, and the result do not really identify the Lean concept as the primary determinant of the study‘s finding. The author should reconsider the role of the Lean concept in the study, especially considering that it is a concept originally derived from manufacturing industry. If the Lean process is still the primary aim of the study, it should be better justified.

The conclusion is more of a hypothesis and future direction instead of conclusion from the study and is finding.

Minor issues

The author should make clear distinction of different components of HRM throughout the manuscript and weather the finding from a survey question is applicable to some or all components. Based on the manuscript, a reader will expect to see at least two unique components being discussed, namely the business management and patient management component.

Data presentation on Table 3 should include the mean differences and its 95%CI, as well as including the p-value in the table (maybe 3 decimal points is required). Should also provide information about the nature of the number, e.g. is it the mean from 5-point Likert scale, 7-point Likert scale, or 10 point numerical rating scale? Is higher number represent greater perceived knowledge?

Given that there is 98 subject in your study, there is not much added value in reported the percentage to 2 decimal points.

May consider performing sub-analysis of data comparing those with prior managerial experience or extensive work experience compared to those with limited experience.

May consider discussing how much HRM is required within a pharmacy curriculum before determining a cutoff point beyond which we should advise an individual to undertake additional qualification such as an MBA degree.

The study reported a 100% response rate, which is very rare in voluntary survey. The author may want to discuss the reasons for this high response rate, e.g. students are asked to complete the survey during a class, and how it may potentially influence the finding, including the influence of perceived power imbalance.
